

# A late-surviving apatemyid (Mammalia: Apatotheria) from the latest Oligocene of Florida, USA

Nicholas J. Czaplewski[1] and Gary S. Morgan[2]

[1] Department of Vertebrate Paleontology, Oklahoma Museum of Natural History, Norman, OK, USA
[2] New Mexico Museum of Natural History, Albuquerque, NM, USA

## ABSTRACT

A new species of Apatemyidae, *Sinclairella simplicidens*, is based on four isolated teeth that were screenwashed from fissure fillings at the late Oligocene Buda locality, Alachua County, Florida. Compared to its only congener *Sinclairella dakotensis*, the new species is characterized by upper molars with more simplified crowns, with the near absence of labial shelves and stylar cusps except for a strong parastyle on M1, loss of paracrista and paraconule on M2 (paraconule retained but weak on M1), lack of anterior cingulum on M1–M3, straighter centrocristae, smaller hypocone on M1 and M2, larger hypocone on M3, distal edge of M2 continuous from hypocone to postmetacrista supporting a large posterior basin, and with different tooth proportions in which M2 is the smallest rather than the largest molar in the toothrow. The relatively rare and poorly-known family Apatemyidae has a long temporal range in North America from the late Paleocene (early Tiffanian) to early Oligocene (early Arikareean). The new species from Florida significantly extends this temporal range by roughly 5 Ma to the end of the Paleogene near the Oligocene-Miocene boundary (from early Arikareean, Ar1, to late Arikareean, Ar3), and greatly extends the geographic range of the family into eastern North America some 10° of latitude farther south and 20° of longitude farther east (about 2,200 km farther southeast) than previously known. This late occurrence probably represents a retreat of this subtropically adapted family into the Gulf Coastal Plain subtropical province at the end of the Paleogene and perhaps the end of the apatemyid lineage in North America.

Corresponding author
Nicholas J. Czaplewski,
nczaplewski@ou.edu

## INTRODUCTION

Apatemyids are a unique family of small mammals that are uncommon in Paleogene localities of North America and Europe. Their cranial and postcranial anatomy are known in North America from Paleocene specimens of *Labidolemur kayi* (*Bloch & Boyer, 2001*; *Silcox et al., 2010*) and an Eocene specimen of *Apatemys chardini* (*Von Koenigswald et al., 2005a*), while the unusual dental, jaw, and hand adaptations of these and a few skeletons from Europe indicate that apatemyids evolved convergently with living marsupials in the genus *Dactylopsila* (Petauridae; striped or long-fingered possums) of New Guinea and northeastern Australia, and with the living primate *Daubentonia* (Daubentoniidae;

aye-aye) of Madagascar (*McKenna, 1963*; *West, 1973*; *Von Koenigswald & Schierning, 1987*; *Von Koenigswald et al., 2005a*; *Von Koenigswald et al., 2005b*; *Rose, 2006*). Aspects of apatemyids' ecology, too, were probably similar to these living mammals as well as possibly that of the extinct Oligocene-Miocene metatherian *Yalkaparidon* of Australia (*Beck, 2009*). Like striped possums and the aye-aye, the striped possum-sized apatemyids are presumed (1) to have been arboreal and insectivorous, (2) to have used their pincer-like upper and lower incisors to tear loose bark from trees or rip open rotting wood that might harbor wood-boring insects, and (3) to have used two elongated manual digits to help find and extract the insects from cavities in the wood. The overall time range of the Apatemyidae as presently known spans virtually the entire Paleogene, from the early Paleocene to Eocene in Europe and from the early Paleocene to the late Oligocene (early Arikareean) in North America (*McKenna & Bell, 1997*; *Rose, 2006*; *Gunnell et al., 2008*). Apatemyids are unknown from other continents. The earliest North American genera in this poorly known family are *Jepsenella* and *Unuchinia*. *Jepsenella* is known from the late Torrejonian, *Unuchinia* from the Torrejonian and Tiffanian (Paleocene), and *Labidolemur* from the early Tiffanian. *West (1973)* reviewed the North American Eocene and Oligocene Apatemyidae (not the Paleocene taxa) and reduced several previously named genera (*Apatemys*, *Labidolemur*, *Stehlinius*, *Stehlinella*, *Teilhardella*, and *Sinclairella*) and species to just two genera: *Apatemys*, with two highly variable species, and *Sinclairella*, with a single species. So defined, *Apatemys* has a long temporal range of perhaps 20 million years from the late Paleocene (Tiffanian) until the late middle Eocene (Duchesnean) in North America. All known occurrences of this family in North America are in the western interior or Pacific coastal parts of the continent, in Saskatchewan, Canada, and the states of Montana, Wyoming, South Dakota, Nebraska, Utah, Colorado, New Mexico, and California (*West, 1973*; *Storer, 1996*). Some authors (e.g., *McKenna & Bell, 1997*) followed *West*'s (*1973*) synonymy, but *Russell et al. (1979)* asserted that *Labidolemur* should be maintained as a separate genus for the species *L. kayi*. *Gingerich & Rose (1982)* and *Gingerich (1982)* also retained *Labidolemur* for other Paleocene species. *Rose (2006)* recognized Apatemyidae as the only family within the order Apatotheria, containing six genera, *Unuchinia*, *Jepsenella*, *Labidolemur*, *Apatemys*, *Heterohyus*, and *Sinclairella*. We follow *Rose*'s (*2006*) taxonomy herein but also recognize in addition the recently described European genus *Carcinella* (*Von Koenigswald, Ruf & Gingerich, 2009*).

The latest known apatemyid is *Sinclairella dakotensis Jepsen, 1934*, the only recognized member of its genus. *Sinclairella* occurred in the Duchesnean (late middle Eocene), Chadronian (late Eocene), Orellan (early Oligocene), Whitneyan (early Oligocene) and early Arikareean (early Oligocene) of western North America. Two of the earliest records are identified only as *Sinclairella* sp. (*Storer, 1995*; *Storer, 1996*); later occurrences are published as *Sinclairella dakotensis*. Ages and localities of occurrence (Fig. 1; ages here are adjusted to reflect those provided by *Albright et al., 2008*; *Janis, Gunnell & Uhen, 2008*: appendix 1) are: Duchesnean: Lac Pelletier Lower fauna, Saskatchewan (*Storer, 1995*); Chadronian: Calf Creek Local Fauna (LF), Saskatchewan (*Storer, 1996*), Flagstaff Rim area and Bates Hole, Wyoming (*Emry, 1973*; *West, 1973*), the type locality at Big Corral Draw, South Dakota (*Jepsen, 1934*), Medicine Pole Hills Local Fauna, North Dakota (*Pearson &*

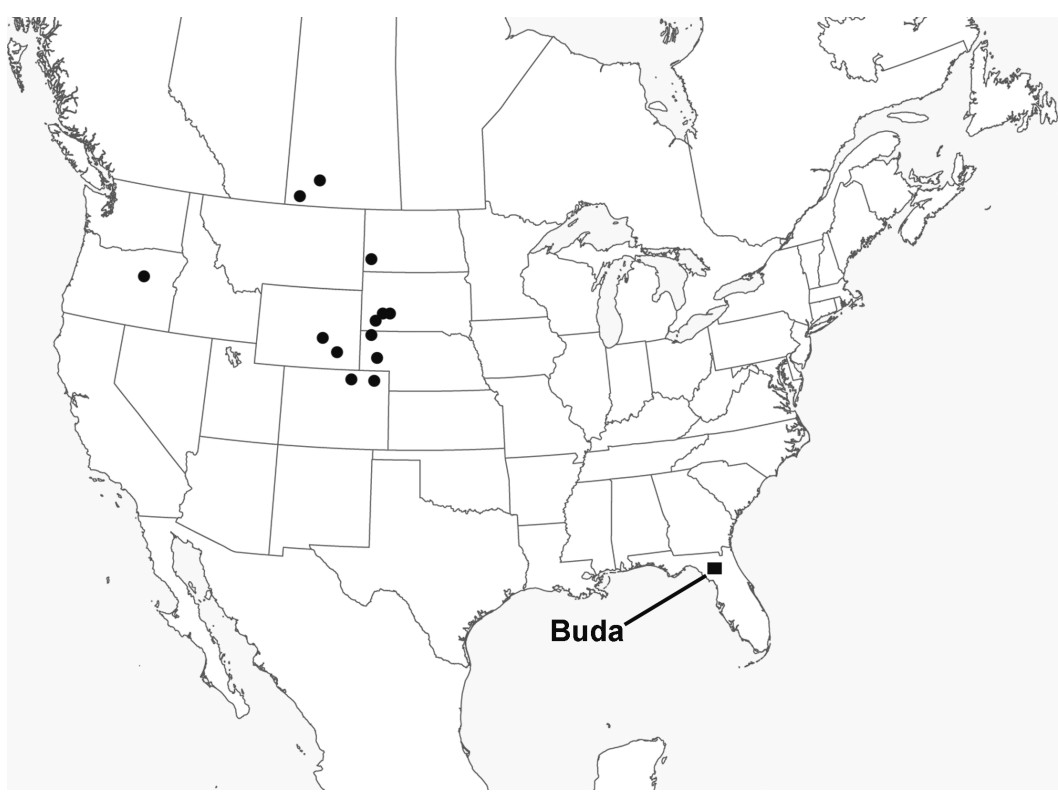

**Figure 1** **Map of North America showing the localities in which the apatemyid *Sinclairella* is known.** Dots represent occurrences of *Sinclairella* sp. and *S. dakotensis*; square represents *Sinclairella simplicidens*. See text for a listing of localities other than Buda, Florida.

*Hoganson, 1995*), Raben Ranch and Twin Buttes Local Faunas, and harvester ant mounds in Sioux County, Nebraska (*Clemens, 1964*; *Ostrander, 1985*; *Ostrander, 1987*); Orellan: Logan County and Weld County, Colorado (*Clemens, 1964*), and Shannon County, South Dakota (*West, 1973*); Whitneyan-early Arikareean: Wh2-Ar1, Harris Ranch badlands, Fall River County, South Dakota (as *Sinclairella* cf. *S. dakotensis*; *Simpson, 1985*); Ar1, Gering Fauna, Nebraska (*Gunnell et al., 2008*); Ar1, Unit C of Turtle Cove Member, John Day Formation, Oregon (29.75–28.8 Ma; *Cavin & Samuels, 2012*). Thus, through *Sinclairella dakotensis*, the North American apatemyid lineage continued into the earliest Arikareean (Ar1).

We report the occurrence of the apatemyid lineage for the first time in the eastern half of North America, and about 5 million years later than the previous latest known occurrences. The occurrence consists of four unassociated, isolated teeth of an apatemyid similar to *Sinclairella dakotensis* that were found in the Buda locality, Florida, which yielded a mammalian fauna of latest Oligocene age (early late Arikareean, Ar3). The Florida *Sinclairella* differs in size and qualitative dental characters from *S. dakotensis*, and clearly represents a distinct new species.

Herein we use the revised chronostratigraphy and biochronology of the Arikareean NALMA of *Albright et al. (2008)* as recorded in the John Day Formation of Oregon. These authors placed the base of the Arikareean NALMA and subbiochron Ar1 at

approximately 30 Ma, with basal temporal boundaries of the other subbiochron intervals as follows: Ar2, 28 Ma; Ar3, 26 Ma; Ar4, 23 Ma; base of Hemingfordian NALMA and He1 subbiochron, 18.5 Ma. Of particular significance to the Buda Local Fauna, *Albright et al. (2008)* established the boundary between the Ar2 and Ar3 at about 26 Ma (late Oligocene), whereas *Tedford et al. (2004)* placed this same boundary at about 23 Ma (earliest Miocene). We also use the broader terms, early Arikareean (Ar1 and Ar2), including the earliest Arikareean (Ar1; 30–28 Ma) and late early Arikareean (Ar2; 28–26 Ma), and late Arikareean (Ar3 and Ar4), including the early late Arikareean (Ar3; 26–23 Ma) and latest Arikareean (Ar4; 23-18.5 Ma). *Rincon et al. (2015)* recently subdivided the Arikareean somewhat differently, still recognizing the Ar1–Ar4 subbiochrons with the same boundaries as in *Albright et al. (2008)*, but using the term early Arikareean only for Ar1 faunas, middle Arikareean for Ar2, Ar3, and earliest Ar4 faunas, and late Arikareean for most Ar4 faunas.

## Locality and mammalian biochronology

The Buda Local Fauna (LF) was found in the Buda Quarry, an abandoned limestone mine located about 8 km southwest of High Springs, Alachua County, northern peninsular Florida (29°45′N, 82°38′W). The Buda LF, discovered in February 1965 by SD Webb, N Tessman, JS Waldrop, and E Kayworth, occurred in a clay-filled sinkhole formed in Eocene marine limestone of the Crystal River Formation. The fossiliferous sinkhole deposit, long since destroyed by mining operations, consisted of three shallow vertical chambers from 1 to 3 m in diameter that probably shared a common opening. The remains of large mammals from the Buda Quarry were found mostly in clays in spoil piles, whereas the small mammals were screenwashed from pockets of clayey sand. The fossils occurred primarily as isolated teeth and disarticulated postcranial bones (*Frailey, 1979*).

About 20 species of mammals are presently recognized in the Buda fauna (Table 1). *Frailey (1979)* reported 12 taxa of large mammals: five carnivores (the amphicyonid *Daphoenodon notionastes* and the canid *Bassariscops achoros*, both described as new species, the canid *Cynarctoides* sp., a mustelid, and a nimravid); two perissodactyls (the small chalicothere *Moropus* sp. and an indeterminate equid); and five artiodactyls (the tayassuid *Cynorca*, a phenacocoeline oreodont, two unidentified camelids, and the hypertragulid *Nanotragulus loomisi*). The most abundant mammal at Buda is the tiny artiodactyl *Nanotragulus loomisi*.

In a systematic review of the borophagine canids, *Wang, Tedford & Taylor (1999)* reexamined the canid sample from Buda, and reassigned several specimens to different taxa than in *Frailey (1979)*. *Wang, Tedford & Taylor (1999)* recognized three small borophagine canids from Buda: *Phlaocyon achoros* (referred to *Bassariscops* by *Frailey, 1979*), *Cynarctoides lemur* (including some specimens originally referred to *Bassariscops achoros* by *Frailey, 1979*), and *Cormocyon* cf. *C. copei*. *Phlaocyon achoros* is known only from Buda, the type locality, but this species is similar to *P. multicuspus* from the late Arikareean of Wyoming (*Wang, Tedford & Taylor, 1999*) and *P. taylori* from the early late Arikareean (Ar3) Brooksville 2 LF in Florida (*Hayes, 2000*). *Cynarctoides lemur* is primarily an early Arikareean species, including samples from the John Day Formation in Oregon and the Sharps Formation in South Dakota. *Wang, Tedford & Taylor (1999)* tentatively referred a
**Table 1** Mammalian members of the Buda local fauna (from *Frailey, 1979*; *Albright, 1998*; *Hayes, 2000*; and G Morgan, pers. obs., 2015).

| Order | Family | Genus and species |
|---|---|---|
| Apatotheria | Apatemyidae | *Sinclairella simplicidens* |
| Rodentia | Jimomyidae | *Texomys* sp. |
| | Heteromyidae | *Proheteromys* sp. |
| | Eomyidae | *Arikareeomys* sp. |
| Soricomorpha | Geolabididae | *Centetodon* cf. *C. magnus* |
| Erinaceomorpha | Erinaceidae | *Parvericius* sp. |
| Chiroptera | Emballonuridae | Undescribed genus and species |
| Carnivora | Canidae (Borophaginae) | *Phlaocyon achoros* |
| | | *Cynarctoides lemur* |
| | | *Cormocyon* cf. *C. copei* |
| | Amphicyonidae | *Daphoenodon notionastes* |
| | Mustelidae | Genus indet. |
| | Nimravidae | Genus indet. |
| Perissodactyla | Equidae | Genus indet. |
| | Chalicotheriidae | *Moropus* cf. *M. oregonensis* |
| Artiodactyla | Tayassuidae | *Cynorca* sp. |
| | Merycoidodontidae (Phenacocoelinae) | Genus indet. |
| | Camelidae | *Nothokemas* sp. |
| | | Genus indet. |
| | Hypertragulidae | *Nanotragulus loomisi* |

single isolated tooth from Buda to *Cormocyon copei*, a species best known from the early Arikareean of Oregon. In a paper reviewing the earliest North American chalicotheres, *Coombs et al. (2001)* referred the small Buda chalicothere to *Moropus* cf. *M. oregonensis*, a species also known from the early late Arikareean (Ar3) of Oregon and Texas.

Buda also yielded a significant small mammal fauna that remains mostly unstudied. *Rich & Patton (1975)* reported an isolated lower molar of the erinaceine hedgehog *Amphechinus* from Buda, which *Hayes (2000)* reidentified as the erinaceid *Parvericius*. Other small mammals from Buda include the new apatemyid described herein, the geolabidid insectivore *Centetodon* cf. *C. magnus*, a large species of emballonurid bat, and three rodents, the jimomyid *Texomys*, the large eomyid *Arikareeomys*, and the heteromyid *Proheteromys*. The two teeth of *Centetodon* from Buda represent one of the youngest known records of this genus. *Korth (1992)* described the new species *Centetodon divaricatus* from the McCann Canyon LF in Nebraska, which he placed in the early late Arikareean (Ar3). However, *Tedford et al. (1996)* and *Tedford et al. (2004)* considered the McCann Canyon LF to be late early Arikareean (Ar2), equivalent in age to faunas from the Monroe Creek Formation. The youngest previous record of *Centetodon* is based on two early Arikareean specimens of *C. magnus* from the Monroe Creek Formation in the Wounded Knee area of South Dakota (*Lillegraven, McKenna & Krishtalka, 1981*). *Korth (1992)* also described a new genus and species of large eomyid rodent, *Arikareeomys skinneri*, from McCann Canyon. The only other records of *Arikareeomys* are represented by an undescribed species from four Florida Arikareean faunas, Buda, Cow House Slough, SB-1A/Live Oak, and White Springs (*Morgan, 1993*; *MacFadden & Morgan, 2003*). *Texomys* occurs in both Buda and the correlative Ar3 Toledo Bend LF in easternmost Texas. These are apparently the

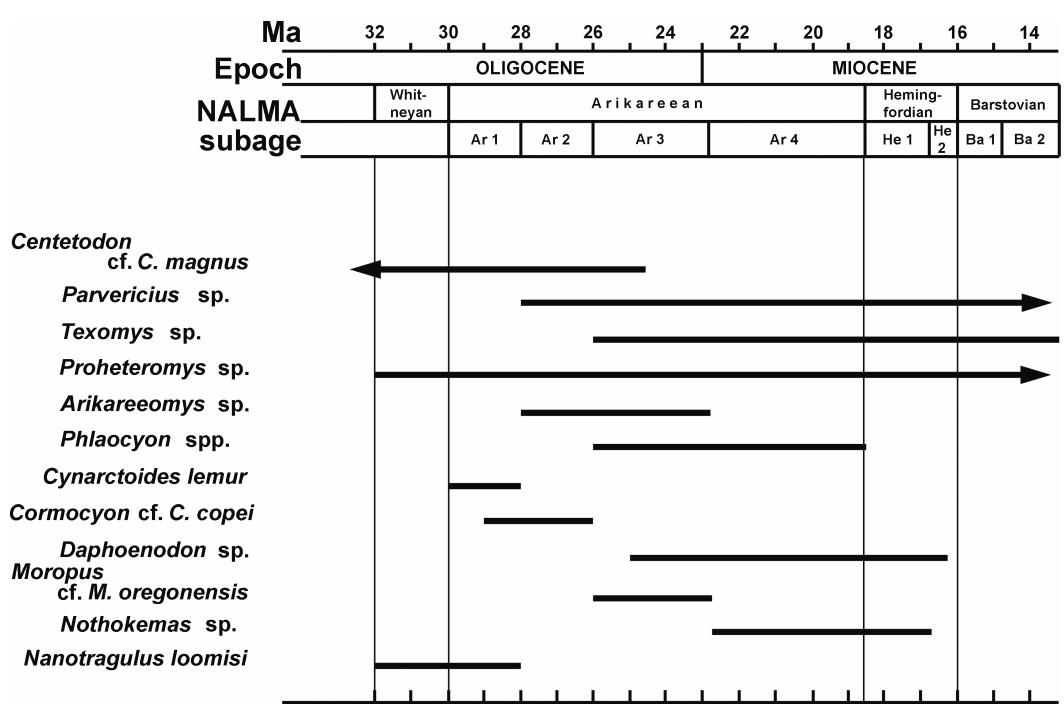

**Figure 2** **Biochronology of Miocene mammal taxa other than *Sinclairella simplicidens* occurring in the Buda Local fauna, Florida, USA, as these taxa are known from North America.** The chart does not include their Buda occurrences.

only Arikareean records of *Texomys*, which is otherwise known from the Hemingfordian of Panama and the Barstovian of Texas and Louisiana (*Slaughter, 1981*; *Albright, 1996*; *MacFadden et al., 2014*).

*Frailey (1979)* considered the Buda LF to be late Arikareean in age. *Tedford et al. (1987)* placed this fauna in the early part of the late Arikareean (Ar3 of *Woodburne & Swisher, 1995*). In comparisons with other Arikareean faunas from Florida and Texas, *Albright (1998)* and *Hayes (2000)* agreed with *Tedford et al. (1987)* in placing the Buda LF in the early late Arikareean (Ar3, 24-22 Ma). *Tedford et al. (2004)* gave the same approximate age range for Buda (24-23 Ma), but placed this fauna in the late early Arikareean (late Ar2). Based on the revised age ranges for the subdivisions (subbiochrons) of the Arikareean NALMA presented by *Albright et al. (2008)*, the Buda LF would be placed in the early late Arikareean (late Ar3; 24-23 Ma; latest Oligocene) (Fig. 2).

Although a number of mammals from the Buda LF are age-diagnostic, depending upon the genus or species, they seem to provide somewhat conflicting evidence pertaining to the site's age. Some mammals from Buda suggest a late early Arikareean age (Ar2), whereas other taxa are indicative of early late Arikareean faunas (Ar3). Furthermore, most of the previous papers that have discussed the mammalian biochronology of Buda (*Frailey, 1979*; *Tedford et al., 1987*; *Tedford et al., 2004*; *Albright, 1998*; *Hayes, 2000*) were written prior to the significant changes in the boundaries for the subbiochrons of the Arikareean NALMA proposed by *Albright et al. (2008)*. Mammals from Buda indicating an Ar3 age are *Moropus*, *Daphoenodon*, and *Texomys*. *Moropus* and *Daphoenodon* make their first appearance in the

John Day sequence at approximately the same time, in the Ar3 at about 25 Ma (*Albright et al., 2008*). *Texomys* does not occur in the John Day Formation, but all other records of this genus are from late Arikareean (Ar3) or younger faunas (*Albright, 1996*). As noted above, *Texomys* has a southern distribution, including Florida, Louisiana, Texas, and Panama, and the earliest occurrences of this genus are in Buda and the correlative Ar3 Toledo Bend LF on the Gulf Coastal Plain of Texas (*Albright, 1996*).

Mammals from Buda indicative of an early Arikareean (Ar1/Ar2) age are the new apatemyid, *Centetodon, Cynarctoides lemur, Cormocyon* cf. *C. copei, Arikareeomys*, and *Nanotragulus loomisi*. Excluding Buda, apatemyids are otherwise unknown after the earliest Arikareean (Ar1) in North America (*West, 1973*). The youngest records of *Centetodon* are from the Wounded Knee Fauna of South Dakota (*Lillegraven, McKenna & Krishtalka, 1981*) and the McCann Canyon LF of Nebraska (*Korth, 1992*), both of which are correlatives of the Monroe Creek Fauna of late early Arikareean age (Ar2). *Wang, Tedford & Taylor (1999)* referred specimens of borophagine canids from Buda to *Cynarctoides lemur* and *Cormocyon* cf. *C. copei*, species originally described from the early Arikareean (Ar1/Ar2) Turtle Cove Member of the John Day Formation in Oregon. The fauna from the type locality for *Arikareeomys*, the McCann Canyon LF in Nebraska, was considered a correlative of the late early Arikareean (Ar2) Monroe Creek Fauna by *Tedford et al. (1996)* and *Tedford et al. (2004)*. In addition to Buda, the three other faunas from Florida in which *Arikareeomys* occurs, Cow House Slough, White Springs, and SB-1A/Live Oak, have been considered late early Arikareean in age (*Frailey, 1978; Morgan, 1993; Hayes, 2000; MacFadden & Morgan, 2003*). However, these three faunas, like Buda, would now be considered early late Arikareean (Ar3) based on the placement of the lower boundary for the Ar3 at 26 Ma (*Albright et al., 2008*). The tiny ruminant *Nanotragulus loomisi* from Buda, as well as the Brooksville 2 LF in Florida, is most similar to samples of *N. loomisi* from early Arikareean faunas in the northern Great Plains (*Frailey, 1979; Hayes, 2000*).

*Albright (1996); Albright (1998)* and *Albright (1999)* considered Buda and the Toledo Bend LF in easternmost Texas to be correlative early late Arikareean (Ar3) faunas based on the co-occurrence of *Daphoenodon notionastes, Moropus*, and *Texomys*. However, these two faunas do have different species of *Nanotragulus*. Buda has the small early Arikareean species *N. loomisi*, whereas Toledo Bend has a larger species, similar to *N. ordinatus* and *N. matthewi*, that is more common in the late Arikareean (*Frailey, 1979; Albright, 1999; Hayes, 2000*).

The conflicting evidence from certain taxa indicating a late early Arikareean (Ar2) age and other taxa indicating an early late Arikareean (Ar3) age for Buda has led several workers to regard Buda and other Florida Arikareean faunas (e.g., Brooksville 2) as "medial Arikareean," essentially combining Ar2 and Ar3 (e.g., *Albright, 1998; Wang, Tedford & Taylor, 1999; Hayes, 2000*). *Rincon et al. (2015)* used the term "middle Arikareean" for the time interval between about 28 and 22 Ma, encompassing the Ar2, Ar3, and early part of Ar4. The presence of *Centetodon* cf. *C. magnus*, the John Day canids *Cynarctoides lemur* and *Cormocyon* cf. *C. copei, Arikareeomys*, and *Nanotragulus loomisi* provides the most compelling biochronologic evidence for a late early Arikareean (Ar2) age for the Buda LF. None of these species occurs after the early Arikareean in western faunas,

and *Sinclairella* is otherwise unknown in faunas younger than earliest Arikareean (Ar1). However, *Dapheonodon* and *Moropus* are unknown from early Arikareean faunas, with their earliest well-dated appearance in the John Day Formation in Oregon in the early late Arikareean (Ar3) at about 25 Ma (*Albright et al., 2008*). Together with the records from the John Day Formation and the Toledo Bend LF in Texas (*Albright, 1998*; *Albright, 1999*; *Albright et al., 2008*), Buda appears to represent one of the earliest occurrences of both *Daphoenodon* and *Moropus*. Pending a more detailed biochronology of Florida Arikareean faunas, we tentatively regard the Buda LF as early late Arikareean in age (Ar3; 25-23 Ma).

## METHODS AND MATERIALS

The electronic version of this article in Portable Document Format (PDF) will represent a published work according to the International Commission on Zoological Nomenclature (ICZN), and hence the new names contained in the electronic version are effectively published under that Code from the electronic edition alone. This published work and the nomenclatural acts it contains have been registered in ZooBank, the online registration system for the ICZN. The ZooBank LSIDs (Life Science Identifiers) can be resolved and the associated information viewed through any standard web browser by appending the LSID to the prefix http://zoobank.org/. The LSID for this publication is: urn:lsid:zoobank.org:pub:76E08309-BD0A-4665-AA39-D36450AAE5FF. The online version of this work is archived and available from the following digital repositories: PeerJ, PubMed Central and CLOCKSS.

Dental terminology follows that used by *Van Valen (1966)* and *Silcox et al. (2010)*. Specimens were measured and photographed at the Oklahoma Museum of Natural History, University of Oklahoma, using an Olympus SZX9 stereomicroscope with an eyepiece reticle, and a Zeiss digital scanning electron microscope, respectively. Anteroposterior length (APL) was the greatest length measured anteroposteriorly along the labial side of the tooth; transverse width (TW) was the greatest width of the tooth measured transversely from lingualmost extent to the labialmost extent. Additional tooth measurements were made in a manner following *Von Koenigswald, Ruf & Gingerich* (*2009*: Table 4): Width 2 was the diagonal width from the lingual base of the protocone to the edge of the tooth labial to the paracone; Width 3 was the diagonal width from the lingual base of he protocone to the edge of the tooth labial to the metacone; Width 4 was the diagonal width from the posterolingual corner of the talon at the hypocone to the anterolabial corner of the tooth at the parastyle.

## RESULTS

### Systematic paleontology

Order APATOTHERIA *Scott & Jepsen, 1936*
Family APATEMYIDAE *Matthew, 1909*
*SINCLAIRELLA Jepsen, 1934*
*SINCLAIRELLA SIMPLICIDENS*, new species.

Holotype. UF 97383, right M2 (Fig. 3).

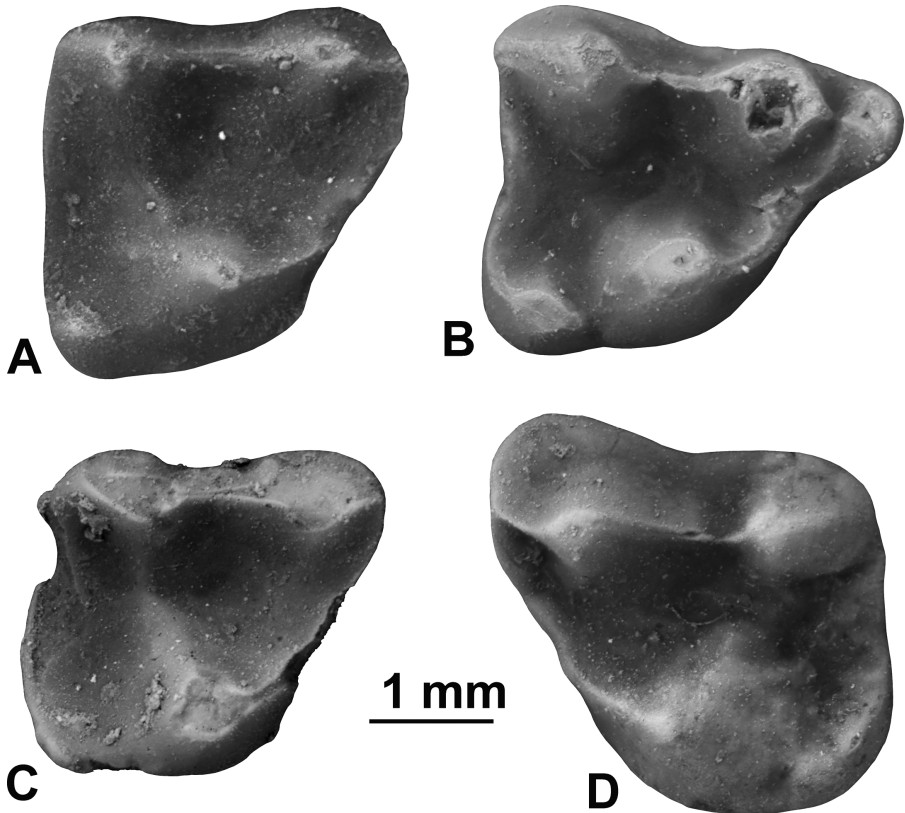

**Figure 3** **Upper molars of *Sinclairella simplicidens* n. sp.** (A) UF 97383, right M2 (holotype) in occlusal view; (B) UF 97385, right M1 in occlusal view; (C) UF 97384, damaged right M2 in occlusal view; (D) UF 97382, left M3.

Hypodigm. Type and UF 97385, right M1; UF 97384, damaged right M2; UF 97382, left M3.

Locality and Horizon. Buda locality, Alachua County, Florida. Fissure fillings in solution cavities formed within the Crystal River Formation, early late Arikareean (Ar3) land mammal age, latest Oligocene (see 'Locality and Mammalian Biochronology').

Etymology. *simplicis*, Latin, "simple;" *dens*, Latin, "tooth;" in reference to the simplicity of the crowns of the upper molars.

Diagnosis. Upper molars simpler than in *Sinclairella dakotensis* and all other known apatemyids in that they lack conules except for a weak paraconule on M1, anterior cingula are absent, labial stylar cusps are absent except for a strong parastyle on M1, and stylar shelves (labial cingula) are virtually absent on all three upper molars, the only remnant being a weak shelf labial to the paracone on M3 that is much less prominent than that on any other known apatemyid. Hypocones on M1 and M2 smaller and on M3 larger than in *S. dakotensis*. Centrocristae (postparacrista and premetacrista) on upper molars straighter than in *S. dakotensis*. M2 relatively small, having about the same occlusal area as M1 and less than M3, about as long as wide, transversely much narrower than the M2 in *S. dakotensis*, and with distal edge continuous from hypocone to postmetacrista supporting a large posterior basin. In size, molars are slightly smaller than in *S. dakotensis* (Table 2).

**Table 2 Measurements (in mm) of upper molars of *Sinclairella dakotensis* from western North America and *Sinclairella simplicidens* n. sp. from the Buda local fauna, Alachua County, Florida.** The measurements of *S. dakotensis* are taken from *West (1973)*.

| Tooth | Measurement | *Sinclairella simplicidens* | *Sinclairella dakotensis* |
|---|---|---|---|
| M1 | APL | 3.36 ($N = 1$) | 3.9–4.1 ($N = 3$) |
| M1 | TW | 2.80 ($N = 1$) | 3.3–3.7 ($N = 3$) |
| M1 | Width 2 | 2.90 ($N = 1$) | |
| M1 | Width 3 | 3.10 ($N = 1$) | |
| M1 | Width 4 | 4.10 ($N = 1$) | |
| M2 | APL | 2.80–2.84 ($N = 2$) | 3.4–3.8 ($N = 4$) |
| M2 | TW | 2.68–3.04 ($N = 2$) | 4.1–4.6 ($N = 4$) |
| M2 | Width 2 | 3.10–3.20 ($N = 2$) | |
| M2 | Width 3 | 2.90–3.10 ($N = 2$) | |
| M2 | Width 4 | 3.70–3.80 ($N = 2$) | |
| M3 | APL | 3.24 ($N = 1$) | 3.2 ($N = 1$) |
| M3 | TW | 3.28 ($N = 1$) | 4.4 ($N = 1$) |
| M3 | Width 2 | 3.60 ($N = 1$) | |
| M3 | Width 3 | 3.20 ($N = 1$) | |
| M3 | Width 4 | 4.20 ($N = 1$) | |

**Notes.**
APL, Anteroposterior length; TW, Transverse width; *N*, Sample size.
Additional measurements as defined by *Von Koenigswald, Ruf & Gingerich (2009)* and in the text are provided for *Sinclairella simplicidens* (Width 2, Width 3, Width 4).

Description. *Sinclairella simplicidens* is known only by the four isolated teeth listed above (Fig. 3). The four teeth represent a minimum of two individuals based on two right M2s; one of these, UF 97384, also shows more wear than the holotype UF 97383. Size is smaller than in *S. dakotensis* and larger than in *Apatemys uintensis*. The molars are brachydont and generally lack sharp crests, although the centrocrista forms a fairly strong ridge between the paracone and metacone of each molar. In occlusal view the centrocrista, preparacrista and postmetacrista form a nearly straight ridge on each upper molar. Paracones and metacones are moderately tall cusps that are oval in occlusal aspect. The molars are simplified by comparison with those of all other known apatemyids.

The M1 is roughly quadrangular in occlusal outline, with a prominent anterior projection formed by the paracone and parastyle. The three main cusps, protocone, paracone, and metacone are roughly equal in size. A smaller hypocone is present on the posterolingual corner of the tooth; it bears weak, short ridges extending anteriorly and posterolabially from its rounded summit. A small, weak paraconule occurs lingual to the paracone. A preprotocrista extends from the protocone anteriorly and labially to the paraconule and parastyle, forming a shelf. A short, notched preparacrista connects the parastyle and paracone. In occlusal view the labial cusps (parastyle-paracone-metacone) are connected by crests (preparacrista-centrocrista-postmetacrista) and aligned nearly straight from anterior to posterior.

Two specimens of M2 are available, the holotype UF 97383 and UF 97384. UF 97384 is somewhat damaged, with minor breakage on its distal edge and preprotocrista, and missing small portions of its enamel on the posterior edge and on the apex of the protocone and labial edge due to breakage or abrasion. The M2 has three main cusps, with the protocone

appearing somewhat larger than the paracone and metacone. A small, low hypocone lends a squarish outline to the posterior half of the tooth. The trigon basin between the three main cusps is relatively shallow and featureless. The hypocone is small and less projecting posteriorly than in *S. dakotensis*. Instead the distal edge of the tooth is nearly straight to barely convex between the hypocone and metacone. A preprotocrista curves forward from the protocone toward the anterior of the paracone, forming a small shelf that closes off the trigon basin anteriorly. There is no parastyle, stylar shelf, or cingula. The preparacrista and centrocrista are aligned nearly straight anterior to posterior in occlusal view; postmetacrista is weak or absent.

The M3 has three main cusps, with the protocone somewhat larger than the subequal paracone and metacone. The hypocone is fairly large, low, rounded, and curved, and is more elongated transversely than in M1 and M2, forming a rounded posterolingual corner on the tooth. As in M1 and M2, the preprotocrista forms a shelf between the protocone and the preparacrista. There is no parastyle. The preparacrista and centrocrista are aligned straight anterior to posterior along the labial half of the tooth; postmetacrista is absent. In occlusal outline, the paracone bulges anteriorly and labially, with a weak labial shelf, but there is no hint of a shelf labial to the metacone. Posterior edge of the M3 is nearly straight, as in M2.

Comparisons. The Buda LF apatemyid consists of four unassociated upper molars representing the loci M1, M2, and M3. The Buda LF specimens can be confidently assigned to the Apatemyidae, whose upper molars are characterized as being relatively low-crowned, transversely rather narrow, bearing three main cusps plus a small hypocone, conules absent or very small, and moderate stylar shelves lacking mesostyles but having strong parastyles (*Rose, 2006*; *Gunnell et al., 2008*). The genus *Sinclairella* is further characterized as being relatively large, with upper molars squarish, lacking ectoflexi, and bearing large hypocones (*West, 1973*). The genus presently includes only one species, *Sinclairella dakotensis*. Given the differences in the few known upper molars of *S. simplicidens* from those of *Sinclairella dakotensis* it is possible that the apatemyid in the Buda LF represents not only a new species but a new genus. Many named apatemyids are based on more complete material, often lower jaws with teeth, skulls, or even complete or partial skeletons. However, until more and better specimens are found and the diagnosis can be expanded with additional characters of the skeleton, skull, or at least the other teeth besides upper molars, it seems parsimonious at present to consider the Florida apatemyid as a new species.

*Silcox et al. (2010)* provided the only available phylogenetic analysis of Apatemyidae, and included *S. dakotensis* in their study. In the characters they developed that can be assessed in the Buda LF molars, two that pertain to the upper molars were found to be synapomorphies of the Apatemyidae: M1 metaconule absent, and M1 paraconule appressed to paracone. In the Buda M1s, metaconules are absent and the paraconule is reduced but situated near the paracone. Several of *Silcox et al.*'s (*2010*) characters pertaining to the upper molars appeared as synapomorphies supporting a node containing *S. dakotensis* and *Heterohyus nanus*, a European taxon. These synapomorphies are: M1 protocone position central on the tooth, M2 hypocone large, similar in size to the protocone, M3 hypocone large, upper molar stylar shelf narrow (buccal cingulum only) or absent, molar cusps relatively
blunt (bunodont). Relative to these character states, the Buda LF specimens resemble *S. dakotensis* and *H. nanus* in sharing the M1 protocone position, narrow stylar shelves, and blunt cusps, but differs from them in having the M2 hypocone distinctly smaller than the protocone, and the M3 hypocone small instead of large. *Sinclairella dakotensis* showed no apomorphies in the upper molars among apatemyids in their study (*Silcox et al., 2010*); however, another European taxon, *Carcinella sigei,* showed an upper molar autapomorphy in having the M1 metaconule and paraconule both weak or absent. This condition, too, differs in the Buda LF M1, which has a weak paraconule and no metaconule.

Fossils of apatemyids are generally rare or uncommon in early Paleogene faunas. When they are encountered, the most commonly found elements are lower jaws or fragments; maxillary fragments and upper molars are quite rare in collections (*West, 1973*; Fig. 4). Thus, the Florida sample is rather unusual in that it consists of isolated upper molars and no lower teeth. When *West (1973)* reviewed all the North American Eocene and Oligocene apatemyids, he studied the type specimen of *S. dakotensis*, YPM PU 13585. The type included the only cranium of the species and was one of the few available specimens including upper molars (*West, 1973*). The holotype YPM PU 13585 was lost in the mail in 1976 and had not been molded before it was loaned (D Brinkman and M Fox, pers. comm., 2001). As a result, our comparisons of *S. simplicidens* were necessarily based on the diagnoses, illustrations, and measurements of *S. dakotensis* upper molars provided in *Jepsen (1934)*, *Clemens (1964)* and *West (1973)*.

The external (labial) shelf in *Sinclairella dakotensis* (and other earlier apatemyids) forms prominent projections anterolabial to the paracone in M2 and M3 and labial to the metacone in M2 (Fig. 4). *Jepsen (1934)* noted that the stylar shelf in M1 supported a number of small cuspules, the largest of which occurred in the metastyle and mesostyle positions (although he commented that these cuspules were not true styles). The labial shelf in M2 is wide and elevated as an irregular ridge along its outer border. In M3 the external cingulum is present anterolabial to the paracone but absent or minute labial to the metacone. *Clemens (1964)* noted no significant differences between Chadronian and Orellan specimens of *S. dakotensis* from the Great Plains and the type specimen of *S. dakotensis*, although he did mention minor differences in details of crown morphology of the upper molars. In a maxillary fragment from Colorado (University of Kansas Museum of Natural History 11210), the M1 has a smooth stylar shelf with no evidence of stylar cusps, unlike the holotype, and the M2 in occlusal view has a stylar shelf represented by anterolabial (with a small parastyle) and posterolabial projections separated by a distinct ectoflexus, and a convex rather than concave posterior margin. Another Orellan specimen (University of Colorado Museum, Boulder, no. 20173), an isolated M2 from Colorado, lacks the stylar cusps in the position of a parastyle and mesostyle (*Clemens, 1964*), but also retains anterolabial and posterolabial projections separated by a deep ectoflexus.

Compared to *S. dakotensis*, the crowns of the upper molars of *S. simplicidens* are even less distinctly cuspate and more simplified, with the bases of the cusps merging into smooth-bottomed valleys between cusps. The crests extending from the major cusps are somewhat better developed. Labial shelves are essentially absent and they completely lack cuspules in *S. simplicidens* compared to *S. dakotensis*. Although the M1 retains a strong

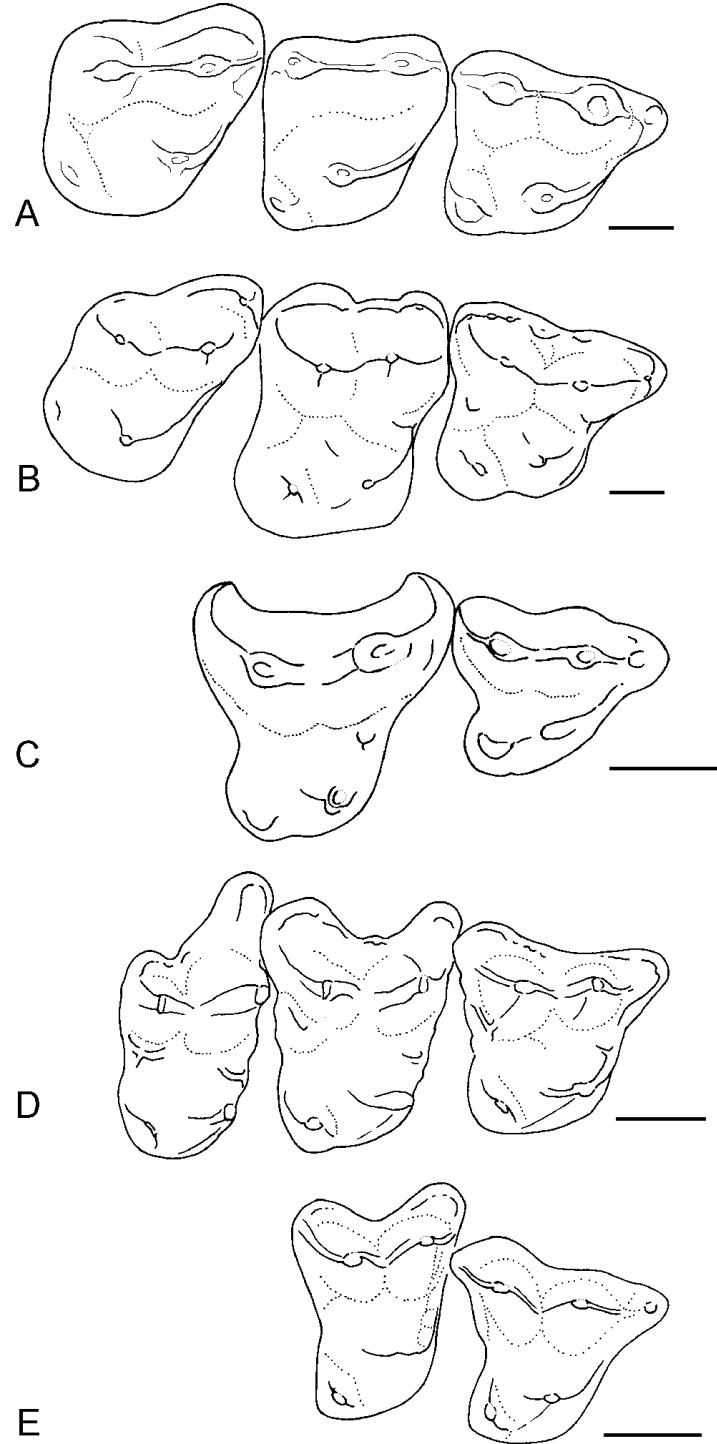

**Figure 4** **Occlusal outlines of upper molars of several North American apatemyids redrawn so that M1s are same size (scale bars at right equal 1 mm).** (A) *Sinclairella simplicidens* n. sp. (composite molar row reconstructed from isolated teeth; see Fig. 2; M3 reversed); (B) *Sinclairella dakotensis* molar row redrawn from *Jepsen* (*1934*; YPM PU 13585; Chadronian); (C) *Stehlinella* (=*Apatemys*) *uintensis* M1–M2 redrawn from *Matthew* (*1921*; AMNH 1903; Uintan); (D) *Apatemys* sp. molar row drawn from *McKenna* (*1963*; USNM 17765; Wasatchian); (E) *Labidolemur kayi* M1–M2 redrawn from *Gingerich & Rose* (*1982*; UMMP 73496; Clarkforkian).

parastyle anterior to the paracone, the only remnant of a labial shelf in *S. simplicidens* is a smooth rounded shelf anterolabial to the paracone in M3. Unlike *S. dakotensis*, parastyles are absent on M2 and M3 in *S. simplicidens*. *Sinclairella simplicidens* lacks other structures that are present in *S. dakotensis*, such as a paraconule on M2, a lingual cingulum anterior to the base of the protocone in M1, and a low ridge posterolingual to the base of the metacone in M1. Relative to the other upper molars, the M2 is the smallest molar in the row in *S. simplicidens*, whereas M2 is the largest upper molar in *S. dakotensis* (Fig. 4). The hypocone on M2 in *S. simplicidens* is much smaller than that in *S. dakotensis*. The hypocone on M3 is present as an elongated ridge in *S. simplicidens*, whereas it is essentially absent in *S. dakotensis*.

## DISCUSSION

With the description of *S. simplicidens* from the latest Oligocene, the overall time range of existence of the Apatemyidae now virtually spans the entire Paleogene (early Paleocene-late Oligocene) in North America and from the early Paleocene to late Eocene in Europe. Apatemyids are generally uncommon in early Paleogene faunas in North America, although, as *West (1973)* noted, they are not quite as rare as *Simpson (1954)* and others had previously believed. This paper records the only known Paleogene occurrence of an apatemyid in eastern North America. The Florida apatemyids were discovered some 10° of latitude farther south and 20° of longitude farther east in North America than previously known.

Given their unusual niche as "mammalian woodpeckers" or ecological equivalents of striped possums or aye-ayes, apatemyids probably required stable, mature forests, where deposition, burial, and preservation conditions are usually poor. They may also have required numerous dead snags or at least dead wood in which to forage, or some other specialized niche or habitat feature of restricted availability that could have limited their abundance. Regarding the less distinctly cuspate and more simplified crowns, and the singularly small M2 proportions relative to M1 and M3 in *S. simplicidens*, these distinctions might point to some unknown difference in the species' trophic biology from *S. dakotensis* and other Apatemyidae. Alternatively, it could represent a phylogenetic trend unique to the North American lineage of which *S. simplicidens* is a part, because there does not seem to be a general trend among North American apatemyids toward simplification of the upper molars.

The Buda fauna preserves certain genera of mammals that are also known in Arikareean faunas of the northern Great Plains and/or the John Day Formation of Oregon (e.g., *Sinclairella, Centetodon, Cormocyon, Cynarctoides, Phlaocyon, Daphoenodon, Arikareeomys, Moropus,* and *Nanotragulus*). At the same time, the Buda fauna and other Florida faunas of middle and late Cenozoic age reflect a provinciality in which the relatively subtropical Gulf Coastal Plain supported a different fauna from the Great Plains and other parts of temperate North America (*Albright, 1998*). The new Florida apatemyid is a significant tie-in both to the Great Plains faunas and the Gulf Coastal Plain endemism noted by several previous authors. *Albright* (*1998*, see also references cited therein) summarized old and new evidence from Arikareean faunas from Texas and Florida indicating an

Arikareean and early Hemingfordian peak in regional endemism of mammals in the subtropical Gulf Coastal Plain. While the mammalian fauna of this region retained strong continuity with the developing grassland fauna in the northern Great Plains, the coastal plain of the Gulf of Mexico provided a refugium for several genera and species of subtropical-tropical forest-adapted mammals. As a Gulf Coastal endemic member of a genus otherwise known only in late Eocene and early Oligocene northern Great Plains faunas, *Sinclairella simplicidens* in the Buda LF of Florida lends additional support to *Albright*'s (*1998*) interpretation of Gulf Coastal Plain faunal endemism in the Arikareean.

Concomitant with retreat into a subtropical forest refugium, *S. simplicidens* might also represent the end of the North American apatemyid lineage unless it or other apatemyids occurred farther south into the tropics of Mexico and Central America. Several recently discovered early Miocene (latest Arikareean and early Hemingfordian) vertebrate faunas from Panama contain a moderate diversity of rodents and a bat, but lack apatemyids (*MacFadden et al., 2014*). The general scarcity of apatemyids in the fossil record and the paucity of Middle American late Oligocene and early Miocene faunas, except those from Panama, will impede confirmation of this possibility until Middle American paleofaunas are much better known.

There are other possible last occurrences of mammalian lineages in the Gulf Coastal Plains faunas, (reviewed by *Albright, 1998*:180; *Albright, 2005*). *Albright (2005)* noted an occurrence of the distinctive primate *Ekgmowechashala* in the early late Arikareean (Ar3) Toledo Bend LF of Texas as a 4-million-year temporal range extension, and a geographic extension from the northern Great Plain to the Gulf Coastal Plain similar to our record of *Sinclairella simplicidens* in Florida. However, *Samuels, Albright & Fremd (2015)* recently questioned the identity of the Toledo Bend specimen as *Ekgmowechashala* and determined that further study of it will be necessary to determine its true identity. In the Buda LF *Centetodon, Texomys*, and *Arikareeomys* probably represent either "holdovers" or Gulf Coast endemics. *Centetodon* and *Sinclairella* are almost certainly holdovers that apparently survived in the Gulf Coastal Plain after their earlier local extinction in the Great Plains. *Texomys* seems to be a Gulf Coastal (and Middle American) endemic because it is currently known only from Panama, Texas, Louisiana, and Florida (*Albright, 1996*; *MacFadden et al., 2014*). *Arikareeomys* is known primarily from Florida (four Arikareean records), with the exception of the type locality—the McCann Canyon LF in Nebraska of *Korth (1992)*.

Gulf Coastal Plain faunas in the Arikareean and early Hemingfordian also reflect an early, temporary expansion of the tropical Middle American fauna into this region (*Webb, 1977*; *MacFadden & Webb, 1982*; *Albright, 1998*). Several bat families with Neotropical affinities in Florida late Oligocene and early Miocene faunas affirm this Neotropical link for at least these volant mammals; the faunas and families include the Buda LF (Emballonuridae) and those from I-75, Brooksville 2, and Thomas Farm (Emballonuridae, Mormoopidae, Speonycteridae, and Natalidae; *Morgan & Czaplewski, 2003*; *Morgan & Czaplewski, 2012*; *Czaplewski & Morgan, 2012*). The bats are not holdovers since none of these families are known in Great Plains Oligocene-early Miocene faunas (nor elsewhere on the continent). However, the Neotropical bats provide further evidence that Florida, and presumably the

remainder of the Gulf Coastal Plain, did provide a subtropical/tropical refugium for certain mammals in the Arikareean and on into the early Hemingfordian.

**Abbreviations**

| | |
|---|---|
| **UF** | Florida Museum of Natural History, University of Florida, Gainesville, Florida, USA |
| **YPM PU** | Yale University Peabody Museum of Natural History-Princeton University, New Haven, Connecticut, USA |
| **NALMA** | North American Land Mammal Age |
| **Ar** | Arikareean North American Land Mammal Age |
| **He** | Hemingfordian North American Land Mammal Age |
| **LF** | Local fauna |
| **Ma** | millions of years before present |
| **APL** | Anteroposterior length |
| **TW** | Transverse width |
| **N** | Sample size |

# ACKNOWLEDGEMENTS

We thank Richard Hulbert and S David Webb, Florida Museum of Natural History, for loaning us specimens and Roger Burkhalter for help in image processing. NJC wishes to thank the Director of the Oklahoma Museum of Natural History for partial support of the research for this project. Ken Rose, Mary Silcox, and Matthew Weiler provided helpful comments on an earlier version of this manuscript.

## Funding

Funds for the completion of this project were received from Department of Vertebrate Paleontology and the Director of the Oklahoma Museum of Natural History. The funders had no role in study design, data collection and analysis, decision to publish, or preparation of the manuscript.

## Grant Disclosures

The following grant information was disclosed by the authors:
Department of Vertebrate Paleontology.
The Director of the Oklahoma Museum of Natural History.

## Competing Interests

The authors declare there are no competing interests.

## Author Contributions

- Nicholas J. Czaplewski conceived and designed the experiments, performed the experiments, analyzed the data, wrote the paper, prepared figures and/or tables, reviewed drafts of the paper.
- Gary S. Morgan conceived and designed the experiments, performed the experiments, analyzed the data, wrote the paper, reviewed drafts of the paper.

## Animal Ethics

The following information was supplied relating to ethical approvals (i.e., approving body and any reference numbers):

The vertebrate involved is a fossil and member of a group that has been extinct for over 20 million years. Thus no institutional review is possible, applicable or necessary.

## Data Availability

The research in this article did not generate any raw data not already included in the article.

## New Species Registration

The following information was supplied regarding the registration of a newly described species:

Sinclairella simplicidens LSID: urn:lsid:zoobank.org:act:E36C99B3-14F3-4D04-86C6-C8CCC3FACFB6

Publication LSID: urn:lsid:zoobank.org:pub:76E08309-BD0A-4665-AA39-D36450AAE5FF.

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
