# Peer review of "A late-surviving apatemyid (Mammalia: Apatotheria) from the latest Oligocene of Florida, USA"

_PeerJ, doi:10.7717/peerj.1509_

## Round 0.1 · original submission · Minor Revisions

You present an interesting discovery, which is well worth being published. The manuscript is in a good state, but some minor additional changes would make it of even higher quality and easier to follow. The main points to address are:

• Add biostratigraphic range zone chart: The biostratigraphic discussion forms an important part of your article, which would be more easy to follow and adding considerably from adding a figure illustrating overlapping and conflicting biostratigraphic ranges of the Buda LC compared with other localities and/or standard zonation (see comments by reviewer 2)

• Improvement of SEM Figures: The contrast in the figure could be improved and holotype should be tilted labially (see comments by reviewer 1)

• Discrepancies between the abstract and the diagnosis: There are some differences in the specific characters used to differentiate the new species between the diagnosis and the abstract, which should be resolved for the sake of consistency (see comments by reviewer 2)

• Cusp diagnosis and measurements: Please add a reference which defines the used cusp nomenclature and briefly describe how you measure the teeth (see comments by reviewer 2)

In addition to the other suggestions from the reviewers, please also attend to the following:

Line 31: “long temporal range in North America (also Europe)”; it is not entirely clear if you mean that the taxon is also known from Europe or that it also has a long temporal range in Europe; please reformulate

Line 44-46: please add a reference for the statement that they evolved convergently with living marsupial and primates

Line 59-60: please add “(Paleocene)” after Torrejonian and Tiffanian for convenience sake for readers not familiar with these North American Mammal Stages

Line 69-72: it might be worthwile to state if you follow the opinion of West (1973) or the opinion of Russell et al. (1979) and/or Gingerich (1982)

Line 97: please replace “chronostratgraphy”by “chronostratigraphy”

Line 102: as you explain the abbreviation on later in the Locality and Mammalian Biochronology section (line 114), so it would be better to write “Buda LF” in full here

Line 293-294: it would add “(see Locality and Mammalian Biochronology Section)” after latest Oligocene for completeness sake

Line 391-392: please consider replacing “virtually the entire Paleogene… in North America” by “virtually spans the entire Paleogene (early Paleocene-late Oligocene) in North America and from the early Paleocene to late Eocene in Europe”

Line 394: what do you mean by Simpson(1954:1) ?

Line 409: please replace “Tertiary” by “Cenozoic” or at least state which epochs are meant by middle and late Tertiary

Line 440: it would replace “extinction” by “local extinction”

Line 624-: please make this into a proper table like table 2

Figure 1: the figure would benefit by using different symbols to plot the distribution of different species of Sinclairella

·

Basic reporting

The writing is generally clear. A bit of re-organizing is needed to adhere to a more standard format with respect to the Systematic Paleontology section, but this should be easy to do. I find the SEM images a bit "flat" looking because of a lack of contrast. Also the holotype (Figure 2, part A) is a bit tilted labially. They are not so bad as to be unpublishable, but they could be improved.

I have put specific suggestions for reorganization etc. on the annotated pdf of the manuscript.

Experimental design

No comments.

Validity of the findings

Generally a well made argument. I am convinced that this material pertains to an apatemyid, and definitely merits naming as a new species.

Additional comments

A very clear, well written paper. Apart from a couple of missing references, and some needed reorganization, very close to ready to be published.

References missing:
Bloch J.I., Boyer, D.M. 2001. Taphonomy of small mammals in freshwater limestones from the Paleocene of the Clarkforkian. University of Michigan Papers on Paleontology 33: 185-198.

von Koenigswald, W., Rose, K.D., Grande, L. Martin, R.D. 2005. First apatemyid skeleton from the Lower Eocene Fossil Butte Member, Wyoming (USA), compared to the European apatemyid from Messel, Germany. Palaeontographica Abt. A 272: 149-169.

·

Basic reporting

I appreciated the in-depth nature of the locality and mammalian biochronology section. Inclusion of a biostratigraphic range zone chart for the Buda LF taxa would enhance this section and allow for a visual representation of the age range extensions and “conflicting evidence pertaining to the site’s age”.

There are discrepancies in the specific characters used to differentiate the new species between the diagnosis and the abstract. For example, in the abstract the loss of the paracrista and paraconule on M2 is mentioned, however, this is not presented in the diagnosis. Additionally, the larger hypocone on M3 is described in the abstract, but not in the diagnosis.

You may want to provide justification for the selection of the M2 (UF 97383) as the type specimen over the M1 which appears to have more defining characters listed in the diagnosis. This may change when the diagnosis/abstract discrepancy is addressed.

In the etymology section, simplicis and dens should be italicized because it is Latin.

In the description section, replace “front to rear” with anterior to posterior (see lines 324, 335, 341). Also, for consistency labial is used throughout the paper except for lines 357, 358, 361, and 377 where “external” is used. My preference would be to continue using labial.

Experimental design

Add a sentence about which specific cusp nomenclature was utilized/followed when describing the features of the teeth (i.e. Van Valen 1966 or West 1973). This would allow a non-specialist a reference to review to ascertain subjective features, such as weakly developed, as well as basic cusp and tooth terminology.

In regards to tooth measurements, equipment was mentioned but how were the specific measurements acquired? For example was A/P width taken along the labial margin of the tooth, or along the paracone/metacone axis, or perpendicular to the transverse axis of the tooth? Was tooth measurement methodology of a previous author followed?

Validity of the findings

I agree with these specimens representing a new species. You may want to consider having additional comments about the M2 size difference (largest tooth in S. dakotensis and smallest tooth in S. simplicidens). Does this represent a trend within the genus or something more (a new genus???)? Do the known specimens of S. dakotensis provide support for a possible trend? You mention that the crowns of the upper molars of S. simplicidens are even less distinctly cuspate and more simplified than S dakotensis. Do you have any speculation on why this might be or if there is a trend within the genus?

---

## Round 0.2 · Minor Revisions

Thank you for implementing my suggestions and those of the reviewers. The manuscript is as good as accepted, but the resolution of Figure 1 and Figure 2 needs to be improved before officially accepting and publication. Currently, they are too pixelated at 100% of its size. Many thanks for your understanding and looking forward to the resubmitted figures.

---

## Round 0.3 · accepted · Accept

Thank you for uploading new versions of the figures.